# Becoming Dad: Expectant Fathers’ Attachment Style and Prenatal Representations of the Unborn Child

**DOI:** 10.3390/children10071187

**Published:** 2023-07-08

**Authors:** Hedvig Svendsrud, Eivor Fredriksen, Vibeke Moe, Lars Smith, Stella Tsotsi, Anne Karin Ullebø, Gro Vatne Brean, Anne Kaasen, Mona Bekkhus

**Affiliations:** 1Department of Psychology, University of Oslo, 0373 Oslo, Norway; hedvig.svendsrud@psykologi.uio.no (H.S.); eivor.fredriksen@psykologi.uio.no (E.F.); vibeke.moe@psykologi.uio.no (V.M.); lrssmth@gmail.com (L.S.); stella.tsotsi@psykologi.uio.no (S.T.); 2Vestfold Hospital Trust, 3103 Tønsberg, Norway; anne.karin.ullebo@siv.no; 3Center for Child and Adolescent Mental Health, Southern and Eastern Norway, 0484 Oslo, Norway; gro.vatne.brean@r-bup.no; 4Department of Nursing and Health Promotion, Oslo Metropolitan University, 0130 Oslo, Norway; annkaa@oslomet.no

**Keywords:** prenatal representations, adult attachment style, bonding, pregnancy, fathers

## Abstract

How expectant fathers think and feel about the unborn child (prenatal representations), has shown associations with fathers’ postnatal parenting behaviors, observed father–infant interactional quality and child cognitive development. There is limited knowledge about fathers’ prenatal representations. The present study examined if fathers’ partner-related attachment styles were related to their prenatal representations of the unborn child. In the “Little in Norway Study”, an ongoing prospective, longitudinal population-based study, 396 expectant fathers completed the Experiences in Close Relationships Scale at enrollment (mean gestational week = 23.76, SD = 4.93), and in gestational weeks 27–35 completed three questions assessing prenatal representations. Correlations of attachment style and prenatal representations were reported using logistic regression analyses. We found that an avoidant attachment style by fathers were predicted to have absent or negative representations on all three items (1) “strongest feeling about the unborn child” (Cl = 1.19–2.73), (2) “thoughts about child personality” (Cl = 1.16–1.87), and (3) “experiences of relationship with the child” (Cl = 1.14–1.75). Father anxious attachment style was not significantly associated with absent or negative prenatal representations. Results suggest that expectant fathers with a partner related avoidant attachment style have an increased risk of having absent or negative prenatal representations of the unborn child.

## 1. Introduction

Fathers in western countries have increased their participation in caregiving over recent decades, with a three- to six-fold increase over one generation [1]. The caregiving behavior of fathers has important and independent (above maternal) effects on child outcomes [2,3,4,5]. The involvement of fathers during pregnancy, as well as their perinatal behavior, is known to benefit maternal health behaviors and fetal outcomes [6], and also affects the quality of later affective and behavioral involvement with their children [7,8]. Higher levels of stress in fathers during pregnancy has been significantly associated with later parasympathetic functioning in infants, after controlling for maternal stress [9]. We typically perceive the transition to becoming a father as occurring at the birth of the child. However, the father–child relationship already begins to develop during pregnancy [10,11]. It can be argued that an expectant father’s thoughts and feelings about himself as a parent, about his unborn child, and about his relationship with the child represent the very beginning of the father–child relationship, and that this psychological reorganization is among the essential tasks that fathers undertake during their entry to parenthood [12]. These thoughts and feelings, or mental representations, are part of the father’s psychological processes during pregnancy, preparing the prospective father to become a caregiver for his child [13]. Dayton, Levendosky, Davidson and Bogat (2010) [14] suggest that a father’s development of mental representations of the unborn child does not require actual experiences with the child, but develops from an underlying capacity to create a relationship with the “imagined” child. Studies investigating how fathers relate to the “imagined” child have shown modest to robust associations with fathers’ postnatal parenting behaviors [15], the observed father–infant interactional quality [16], and the child’s cognitive development [17]. Thus, father’s representations of the unborn child may have lasting associations with his subsequent caregiving and the child’s development. Understanding what may predict the father–child relationship in the prenatal period can contribute to identifying possible developmental pathways of the postnatal father–child relationship. To date, however, there is limited knowledge about such antecedents of fathers’ prenatal representations. The aim of the present study is to gain a better understanding of fathers’ prenatal representations of their unborn children.

### 1.1. Conceptual Definitions of Prenatal Attachment, Bonding, and Representations

In the research literature, various concepts are used when addressing how parents think, feel, and relate to their unborn child. One previously used concept is “Prenatal Attachment”, referring to parental feelings of attachment to the fetus [18]. Researchers have argued that the use of “attachment” in this sense is misleading, as “attachment” theoretically involves a child or a romantic partner seeking care and security from someone who can provide it. Hence, it can be argued that a parent’s prenatal relationship with the child is not “attachment”, but rather reflects the beginning of the parent’s self-representation as a caregiver to this child [19]. Other researchers refer to “bonding”, which reflects the parent’s experience of the emotional tie with their unborn child [20]. Bowlby described how “internal working models” of the self and others develop in early childhood from experiences with important caregivers, and how they have a lasting impact on how we think, feel, and relate to ourselves and other close people [21,22]. In the romantic-attachment literature, the “prototype hypothesis” suggests that internal working models function as a cognitive prototype with persistent effects on subsequent representations and interactions in close relationships [23]. During pregnancy, parental thoughts, feelings, and ways of relating to the unborn child are often referred to as “prenatal representations” [24].

### 1.2. Fathers’ Prenatal Representations

Despite the growing body of research on fathers during infancy, studies focusing on the role of the father during the perinatal period remain scarce [25]. Little is known about the psychological processes fathers undergo during pregnancy [26]. One study, using a community sample, demonstrated that, as in mothers, fathers also develop an emotional connection with the unborn child [27]. In that study, the father’s prenatal representations of their child were assessed in a semi-structured interview, The Working Model of the Child Interview (WMCI), focusing on the “meaning” the child has to the parent [28]. A strong association was found between fathers’ prenatal and postnatal representations of the children. The findings suggest that even though fathers do not experience the fetus physically as do mothers, the father–child relationship evolves prior to experiences with the physically present infant after birth. Another study found that fathers, more often than mothers, showed emotional distance and detachment in their prenatal representations [29]. Alongside narrative interviews, different questionnaires intended to assess prenatal representations are developed for mothers, and then adapted for fathers [16]. Themes that these questionnaires focus on are, for example, ascribing intentions to the unborn child; positive feelings and thoughts about the unborn child; and differentiation between self and the unborn child. Knowledge of which factors contribute to, and how they may contribute to, the development of a father’s prenatal representations is scarce, but it is reasonable to assume that as for mothers, a father’s adult attachment style, and particularly in relation to his partner, may also be associated with the father’s prenatal representations.

### 1.3. Partner-Related Attachment Style and Prenatal Representations

It has been argued that during pregnancy, the mother–child relationship is more strongly influenced by the pregnant women’s current attachment style, measured in relation to her current partner, than by her attachment pattern as a child [19,30]. Elements of how she thinks and feels about herself as a ‘caregiver’ may be more prominent in the romantic relationship, compared to recollections of the care she received many years ago. Walsh et al. (2020) [19] argued that the caregiving system exhibited in relation to a partner guides the formation of the new caregiving relationship with the unborn child. So far, only a few studies have investigated this theory [30,31,32,33,34,35]. Kobayashi et al. (2021) [32] reported that mothers’ partner-related attachment security was associated with the quality of their prenatal representations of their child. There is further robust evidence that parents’ adult attachment styles, irrespective how they are measured, affect later parenting abilities and parent–child relationships [36,37,38].

Brennan, Clark and Shaver (1998) [39] conceptualized adult romantic attachment as a continuum measure of the avoidance of intimacy and the anxiety of abandonment. They argue that secure individuals with lower levels of attachment insecurity show adaptive levels of relationship-related strategies in dealing with stressful situations. Insecure avoidant individuals exhibit the avoidance of intimacy or feelings of dependence in others and use deactivating strategies. Insecure anxious individuals show higher expectations of the loss of love or of receiving insufficient love in close relationships and use hyperactivating strategies to cope with their relationship insecurities [39,40,41]. Ponti, Smorti, Ghinassi, and Thani (2021) [35] found that partner-related attachment avoidance as well as anxiety was associated with a poorer quality of a mother’s affectionate emotional bond and a lower proportion of positive and warm feelings towards the unborn child during pregnancy. As far as we know, only three studies have shown that a mother’s insecure avoidant attachment style is associated with lower scores on their prenatal representations, compared to mothers with secure and anxious attachment styles [19,31,34]. In two studies, an insecure anxious attachment style was found to predict poorer prenatal representations of the unborn child, compared to avoidant and secure attachment styles [30,33]. In two other studies, an insecure anxious attachment style was related to prenatal attachment quality (affective experience when thinking about the child), but not to intensity (time spent in attachment mode/thinking about the child) [33,42]. Røhder and colleagues (2020) [43] have also shown that a mother’s experienced security with the romantic partner was associated with how they related to the child growing inside them. These findings suggest that for mothers, the current attachment relationship with the expectant father is of importance to the formation of the nascent relationship with the unborn child. Hence, a mother’s attachment style in close relationships seems to be related to the development of her prenatal representations.

A father’s working models of attachment formed during their childhood and their own romantic attachment style at the onset of pregnancy may also impact their prenatal representations of the unborn child [32]. Hjelmstedt and Widström (2007) [44] found that a father’s personality trait of showing detachment in relationships was negatively associated with the development of a paternal–fetal relationship. “Detachment” was assessed by a scale developed for measuring personality traits. It seems reasonable that “detachment” and “attachment avoidance” are somewhat overlapping constructs, as they are both described as the avoidance of intimacy or dependence on others in close relationships. In line with this research, Gøbel et al. (2019) [31] investigated the association between a father’s partner-related attachment style and their bonding to the unborn child. Bonding consisted of two scales: one intensity scale, referring to the amount of time spent mentally preoccupied with the fetus, and one quality scale, referring to the emotional experience related to the fetus. Parity and higher attachment avoidance were associated with less intense bonding, but no association was found for bonding quality [31]. The researchers speculated whether fathers with an avoidant attachment style might generally have a more emotionally distant relationship with their partner, which in turn may lead to less time spent talking and thinking about the unborn child.

Indeed, as for mothers, the prenatal period provides a window for promoting the unfolding father–child relationship, with potential long-term benefits for the future father–child relationship and child’s developmental outcomes [45]. Thus, more knowledge is needed on the development of how expectant fathers relate to their unborn child [46].

### 1.4. The Current Study

To date, the potential impact of a father’s partner-related attachment style on their own prenatal representations about the unborn child remains largely understudied. Disturbances in how fathers relate to their unborn child may indicate suboptimal prenatal paternal involvement, a risk for suboptimal postnatal parenting and a need for intervention during pregnancy. Increased knowledge about which factors are related to the quality of fathers’ prenatal representations is important as this knowledge may inform the development of new methods for assessing prenatal relational risk. Tailored intervention practices aimed at supporting the father–child relationship in the perinatal period are needed [47]. Thus, in this study, we ask if a father’s partner-related attachment style measured early in pregnancy is related to their prenatal representations of the unborn child later in pregnancy. We hypothesize that a father’s attachment insecurity and attachment anxiety, as well as attachment avoidance, will be associated with absent or negative thoughts and feelings about the unborn child, the child’s personality, and their prenatal relationship with the child. In line with previous research [31,44], we expect the association to be stronger for fathers with an avoidant attachment style, than for fathers with an anxious attachment style.

In the current study, our main aim is to examine whether a father’s partner-related attachment style (avoidant or anxious), measured at enrollment (mean gestational week 23.76, SD = 4.93) associates with a father’s prenatal representations measured in the third trimester of pregnancy (gestational weeks 27–35). Prenatal representations were, in the main study (LIN), assessed four times during pregnancy, from enrollment until the third trimester. We drew data on prenatal representations from the third trimester assessment, because a father’s feelings of bonding, and the development of prenatal representations of the unborn child, have been found to increase in strength as pregnancy develops [15,48,49,50]. Specifically, the prenatal representations were operationalized as (a) fathers’ emotions about the unborn child; (b) fathers’ prenatal representation of the personality of the unborn child; (c) fathers’ representations of the relationship with the unborn child.

## 2. Materials and Methods

### 2.1. Participants and Procedure

The study is part of an ongoing prospective, cross-disciplinary longitudinal population-based study, Little in Norway (LIN), following infants and their parents from pregnancy and into middle childhood [51]. All pregnant women receiving routine prenatal care at nine public infant health care clinics across Norway were invited by midwives to participate in the LIN study. The population invited to participate was chosen to ensure the representation of a demographic variety and various background conditions. In the Norway public health care system, treatment is free of charge, and used by practically all pregnant women. Initially, 1036 women consented to participate. Their partners were also invited, and 878 men consented to take part. There were no exclusion criteria, but since the questionnaires were in either Norwegian or English, this recruitment process may have excluded pregnant women and/or their partners who spoke neither of these languages. In general, participants showed a higher educational level than the reference population (*p* < 0.001). Further details about the design and procedure of the complete study have been described elsewhere [51].

All 878 fathers answered the questionnaire about partner-related attachment styles as part of the enrollment, while 396 of them answered the questionnaire about prenatal representations in gestational weeks 27–35. The participants’ mean age was 31.9 years, (range 20–56), 99.5% were living with their partner, and 97.3% of the fathers expressed the pregnancy as wanted. Procedures were in accordance with the ethical standards of the institutional and national research committee and with the 1964 Declaration of Helsinki and its later amendments. The study protocol and the assessment procedures were reviewed and approved by the Norwegian Regional Committees for Medical and Health Research Ethics (reference number 2011/560) [51].

### 2.2. Measures

*Partner-related attachment style.* Fathers’ partner-related attachment security was assessed between gestational weeks 12 and 31, (M = 23.76, SD = 4.93) by the Experiences in Close Relationships Scale (ECR) [39]. The ECR Scale is a self-reported measure designed to assess two dimensions of adult attachment styles in close relationships. The two dimensions are avoidance and anxiety, consisting of 18 items each. Fraley (2019) [52] defines attachment anxiety as the extent to which a person feels uncertain about the availability and responsiveness of an attachment figure. Attachment avoidance is defined as the extent to which a person is uncomfortable opening up to others or using them for attachment functions. The scale has been used in several previous studies [36,53]. A Norwegian study of a population-based sample found the psychometric properties of the ECR to be satisfying when used in a Norwegian context [54]. Response options range from 1 (strongly disagree) to 7 (strongly agree), and composite scores are calculated, with higher scores reflecting a greater level of insecure attachment within each relationship’s domain (range 18–126 on each subscale). Mikulincer and Shaver (2005) [55] reported Cronbach’s alpha coefficients for the ECR to be near or above 0.90 and the test–retest coefficients to be between 0.50 and 0.75. Cronbach’s alpha coefficients in the current full sample were 0.88 and 0.89 for the anxiety and avoidance subscales, respectively.

*Paternal Prenatal Representations.* Fathers’ prenatal representations about the unborn child were assessed using a questionnaire developed for the LIN study. Three questions measuring expectant fathers’ thoughts and feelings about the unborn child, their personality and their experienced relationship with the unborn child, were selected. The first question was (1) “Can you describe your strongest feeling about your child, at the moment?” Respondents were asked to choose one of seven alternatives: (a) happiness, (b) joy, (c) sadness, (d) fear, (e) anxiety, (f) hatred, (g) cannot describe it. The six alternatives were coded as ‘0’ positive (i.e., ‘joy’ or ‘happiness’) or ‘1’ = absent or negative (‘cannot describe it yet’, ‘sadness’, ‘anxiety’, ‘fear’, or ‘hatred’). The second question was (2) “Can you describe your perception about your child’s personality, in one word?”. Respondents were asked to choose one of six alternatives: (a) calm, (b) active, (c) harmonious, (d) demanding, (e) angry, (f) do not have any meaning of the child’s personality yet. Answers referred to as positive were coded as ‘0’ (i.e., calm, active, harmonious) or absent or negative, coded as ‘1’ (i.e., demanding, angry, haven’t any meaning of the child’s personality yet). The third question was (3) “Which of the following words describes your relationship to your child best?”. Response categories were (a) close, (b) warm, (c) neutral, (d) distant, (e) hostile, (f) cannot yet describe the relationship. Positive answers were coded as ‘0’ (i.e., ‘close’, ‘warm’) and absent or negative answers as ‘1’ (i.e., ‘neutral’, ‘distant’, ’hostile’, ’cannot yet describe the relationship’).

*Confounding variables.* The following confounding variables were included in the analyses: (a) parity (“no previous children” coded as ‘0’, or “whether the father already had children” as ‘1’), and (b) fathers’ level of education (measured on a 4 point scale, coded as ‘1’ = “nine- or ten years primary school”, ‘2’ = “high school/college”, ‘3’ = “university ‘4’ years”, 4 = “university > 4 years”).

### 2.3. Data Analysis

First, we examined the distributions of fathers’ attachment style and fathers’ prenatal representations, and bivariate associations among all study variables. Because only ECR avoidance correlated with the representational questions, only ECR avoidance was included into the three logistic regression analyses.

One logistic regression analysis was computed for each prenatal representation. The three outcome variables were (1) “Strongest feeling about the child”, (2) “Perception about child’s personality” and (3) “Relationship with the child”. First, ECR attachment avoidance was entered to examine the main effect on prenatal representations, and in the second step we adjusted for two potential confounders: (1) if it was a father’s first child or not; and (2) the father’s level of education. SPSS 28.0 was used for all statistical analyses.

## 3. Results

### 3.1. Descriptives and Bivariate Correlations

Descriptive statistics and bivariate correlations between the key variables are reported in Table 1. Overall, father mean attachment avoidance score was somewhat lower (*M* = 33.20, *SD* = 12.17) than father mean anxiety score (*M* = 38.26, *SD* = 14.52). In addition, 5.5% (N = 22) of fathers reported “negative feelings” or “not able to describe their feelings” on the prenatal-representation question “strongest feeling about infant at the moment”, and 94.5% reported “positive feelings”. However, on the question “Describe infant personality in one word”, 305 fathers (76.6%) reported “positive thoughts about infant personality”, while 93 fathers (23.4%) reported “negative thoughts”, or “do not have any meaning about infant personality, yet”. For the prenatal representation “Which word does best describe your relationship with your infant”, 233 fathers (58.5%) reported “positive feelings”, while 165 fathers (41.5%) reported “negative, neutral” or “cannot yet describe it”. The strength of the relationship between the three prenatal representational questions was significant but small (*r* = 0.20 to 0.27). The strength of the correlations between attachment avoidance and attachment anxiety was moderate (*r* = 0.35). A significant positive correlation was found between partner-related attachment avoidance and all three variables measuring paternal prenatal representations. Fathers’ avoidant attachment style was positively related to absent or negative representations on all three questions assessing prenatal representations (*r* = from 0.15 to 0.17). The correlations were small, but significant, and rather similar in strength. In contrast, a father’s anxious attachment style was not significantly associated with absent or negative prenatal representations. Therefore, we continued by only examining attachment avoidance as a predictor of prenatal representations.

### 3.2. Logistic Regression Analyses of Attachment Avoidance (ECR) and a Father’s Prenatal Representations of the Unborn Child

In the logistic regression analyses, we examined the associations between a father’s attachment style and prenatal representations in three separate analyses (Table 2). First, we examined the main effect of ECR avoidance on prenatal representations, before we adjusted for potential confounders. Results suggest that fathers with higher attachment avoidance had a higher risk of having absent or negative thoughts and feelings about the child on all three measures of prenatal representations. These associations remained significant, but were reduced in strength when adjusted for potential confounders, i.e., the father’s education and parity, measured at the beginning of pregnancy. Having previous children, coded as ‘1’, showed significant associations with fathers’ prenatal representation «strongest feeling about the child», but only marginally significant levels with the other two prenatal representations “child’s personality” and “thoughts and feelings about the relationship”. Thus, already having children increased the likelihood of having absent or negative feelings about the unborn child. One standard deviation increase in ECR avoidance increased the odds of having absent or negative feelings about the unborn child (OR = 1.80, 95% CI [1.19, 2.73]), representations of the unborn child’s personality (OR = 1.48, 95%CI [1.16, 1.87]), and representations of the paternal relationship with the unborn child (OR = 1.42, 95%CI [1.14, 1.75]).

## 4. Discussion

In this study, we hypothesized that a father’s attachment insecurity (anxiety and avoidance), would predict their absent or negative prenatal representations of their unborn child.

First, we discovered that an insecure avoidant attachment style in fathers early in pregnancy was a predictor for absent or negative prenatal perceptions of their child later in pregnancy. Second, a father’s insecure anxious attachment style was not related to any prenatal representations. The results for the avoidant attachment style and the three representational items followed the same pattern, showing small, consistent, and significant correlations. The relevant mechanisms for understanding our findings, including possible mediators and moderators, are discussed below. 

One possible mechanism explaining our results is that the partner-related attachment style and prenatal representations may be expressions of the same underlying construct, as proposed by Bowlby (1979) [21], namely, the internal working models of attachment relationships. Furthermore, the results lend support to the prototype hypothesis, which proposes that working models of attachment influence relationships over the life span, including both romantic relationships and those formed with one’s own children [56]. Previous research supports the prototype hypothesis by demonstrating associations between maternal working models assessed from the recollections of childhood memories and a mother’s prenatal representations of the unborn child [57]. We suggest that our results are an extension of these findings, by showing that a father’s avoidant attachment style in the romantic relationship is associated with absent/negative prenatal representations. Our results add to previous research suggesting that current working models of attachment relationships (partner-related attachment) are associated with new working models of caregiving relationships (a father’s prenatal representations) [31]. Similar connections have been demonstrated for mothers in prior studies [30,33]. Thus, it seems plausible that the formation of a father’s prenatal representations is related to the father’s inner working models, developed from earlier and ongoing experiences in close relationships. An avoidant attachment style in the romantic relationship is characterized by the dampening of emotions and distancing strategies to maintain independence, control and autonomy [58]. Emotional intimacy and closeness are anticipated as undesirable and uncomfortable. We suggest that this tendency of distancing and suppressing emotions, and rather focusing on behavior in the romantic relationship, may create a propensity for a similar pattern to maintain consistency when developing prenatal representations of the child. The prototype hypothesis may also help us to understand why we did not find an association between the anxious attachment style and absent or negative prenatal representations. The anxious attachment style is characterized by a fear of loss and separation, and a need for the assurance of support in close relationships [58]. Anxiously attached individuals are supposed to have no particular issues with feeling close and connected. Consistency in this working model may comprise a wish for closeness, feelings of affection for the child, enhanced imaginations about the child’s personality, and boosted feelings of being connected. This could be one reason that a father’s anxious attachment style was not related to negative or absent prenatal representations in our sample.

Our results may also be understood in light of Belsky’s (1997) [59] proposal that a partner-related attachment style is not only a product of early experiences of receiving care, but also the ongoing experiences of providing care to the romantic partner. Prenatal representations may be early expressions of the parents’ caregiving system [13,19].

Consequently, both the attachment style related to the partner and prenatal representations could be manifestations of the father’s caregiving system.

Another possible factor explaining our findings is related to the perceptions and coping strategies of stress. Experiences of stress, and the need to use strategies to cope with it, may have been prevalent in the fathers during the third trimester of pregnancy. Becoming a father may be experienced as a life-changing and stressful period. The stress could be related to work–family conflict, role changes, identity, marital satisfaction and well-being. For example, a recent study by Penner et al. (2002) [60] found that fathers, more often than mothers, used suppression as a strategy to handle stress (inhibit emotional responses) rather than reappraisal (changing thoughts about the situation in order to change the emotional response). Individuals with secure and more positive internal working models of themself and others may adapt more easily in the transition to parenthood and have more adaptive strategies for coping with stress [61], for example, through reappraisal [62]. 

The diathesis stress model proposes that the two types of attachment insecurity measured in our study constitute a susceptibility for managing stress in ways that can impair intrapersonal and relational functioning [58]. Thus, attachment avoidance and attachment anxiety may influence a father coping with intrapersonal and interpersonal stress during pregnancy. Fathers with a partner-related avoidant attachment style may use distancing strategies to cope with experiences of stress during pregnancy, both in relation to the pregnant mother and to the unborn child, thus distancing themselves from the partner and the unborn child or suppressing thoughts and feelings about them. Anxious fathers’ prenatal stress may escalate fears of separation and loss in relation to the partner and the unborn child, thus increasing proximity and warmth towards the partner and child to achieve reassurance and comfort. Altogether, this can help us understand why we found an avoidant attachment style to be related to absent/negative prenatal representations, and why we did not find a similar association for the anxious attachment style.

Other possible mechanisms explaining this association could be a father’s capacity for prenatal parental reflective functioning (think about the unborn child as a separate individual with experiences, needs, and a developing temperament and personality) [63]. We suggest that a capacity for imagining the unborn child as a separate person, and being curious about the unborn child’s experiences, without feeling stressed, may lead to an easier pathway for developing prenatal representations about the infant’s personality and the relationship with the infant. It may be that for fathers with an avoidant attachment style, the capacity to be curious about the child as a person are difficult to develop if the father’s attachment style leads them to distance themself from the pregnancy and the child.

Previous research has suggested that fathers’ parenting may be more impacted by the quality of the couple relationship, than is the case for mothers [11,64]. According to previous research, fathers tend to feel more disconnected to the pregnancy than mothers do [65]. Thus, fathers seem more dependent on how the mother shares her perceptions and feelings about the unborn child. From this perspective, pregnant mothers can be considered as “gatekeepers” or “gate-openers” for the prenatal father–child relationship [66]. It seems likely that a father’s partner-related attachment style may influence how they physically relate to the mother’s pregnant body and the unborn child within. This may in turn affect a father’s embodied experience of the child, for example, through the father’s perceptions of the child’s movement qualities, and thus, their perceptions of the child’s personality and temperament. It could be hypothesized that expectant fathers with an avoidant attachment style would not just utilize more emotional distancing strategies, but would also be more likely to create a physical distance from the bodies of their pregnant partners compared to fathers with anxious attachment styles.

Our results are in line with another study by Hjelmstedt et al. (2007) [44], which found an association between an expectant father’s personality trait of showing detachment in relationships (e.g., distance preference in interpersonal relations) and a lower level of bonding with the unborn child. Gøbel et al. (2019) [31] found that a father’s attachment avoidance predicted bonding intensity (how much time fathers spent thinking about their unborn child), but not bonding quality (the emotional experience related to the unborn child). Results from this study showed that attachment avoidance correlated with how much time fathers spent thinking about their child, but in contrast to our results, no association between attachment avoidance and fathers’ emotional experience related to their child was found. Results may be difficult to compare in a precise manner, as the assessment methods used to assess prenatal paternal representations in the study of Gøbel et al. (2019), Hjelmstedt et al. (2007), and in our study were not the same.

Our findings suggest that already having children increases the likelihood of having absent or negative feelings about the unborn child. These results are supported by previous research [29]. One possible explanation may be that during late pregnancy, fathers already having children are preoccupied with providing care to those other children, compromising the mental space for thinking and feeling about the unborn child and leading to more distant and underdeveloped representations. For first-time fathers, the emotional intensity of becoming a father may be higher, leading to the development of more positive expectations towards the child and the father–child relationship. Another possible explanation may be that fathers who have already experienced getting to know a child as a little person, implying a variety of experiences and emotions, do have an increased propensity to develop more nuanced prenatal representations than first-time fathers. In our analyses we included absent, neutral and negative representations into one “risk” category. It is possible that fathers already having children and thus having less intense feelings about the unborn child were categorized as at risk in this way.

### 4.1. Strengths and Limitations

To the best of our knowledge, this is the first study to investigate how a father’s partner-related attachment style predicts subdomains of his prenatal representations. Furthermore, our study is population based, longitudinal and has an adequate sample size to fit the analyses. To account for the possible contributions of other variables, we included the father’s education, and if the father had previous children.

Among the limitations in our study is the extension of the results to the general population, as the fathers in our sample had a higher educational level than the reference population, and a very high percentage of the fathers were living with their partners. The results cannot automatically apply to high-risk groups or a clinical population. The study was conducted in a Norwegian context which also compromises its generalizability to other societal and cultural contexts. Another limitation is that we do not know whether mothers’ prenatal representations and attachment style are associated with fathers’ prenatal representations, and if this would have influenced our findings. Prenatal representations in this study were assessed as separate items. This makes it difficult to compare results with findings from studies which used established and validated measures.

The present study did not investigate how latent (moderating or mediating) variables, such as parental mental health, genetics, temperament, stress, involvement in the pregnancy, dyadic coping in the romantic relationship or prenatal parental reflective functioning might have played out. Finally, because the knowledge about antecedents and correlates of fathers’ prenatal representations are very limited, findings need to be replicated.

### 4.2. Future Directions

We support previous research calling for more work to increase knowledge about fathers’ development of prenatal representations [31,67], and predictors of this, as well as possible moderating and mediating factors. There is a need for a systematic review of questionnaires employed for measuring fathers’ prenatal representations, and further validation. We highlight the need for more research on the role of fathers’ experiences of stress during pregnancy, and how this may explain variations in fathers’ psychological preparation to fatherhood. The prenatal assessment of partner-related attachment style, prenatal representations and experiences of parental stress could function as a guide for assessing vulnerabilities in the emerging father–child relationship and expose the need for early intervention. A compromised prenatal father–child relationship may be a marker for postpartum relationship difficulties [26]. Generally, including fathers more actively in prenatal care contexts might be an opportunity for them to develop a representation of the child, provide fathers with information about the developing fetus, and offering a safe space for them to share thought and feelings about the transition to fatherhood. We suggest that future studies should investigate if knowledge about fathers’ adult attachment styles and prenatal representations are useful in developing interventions supporting fathers in a foundational period in life.

## 5. Conclusions

This study indicates that prospective fathers, who have an avoidant attachment style in relation to the expectant mother, are at a higher risk of having absent or negative prenatal perceptions of their unborn child compared to fathers with a secure attachment style related to their partner. An anxious attachment style in fathers was not related to absent or negative prenatal representations about the unborn child. Further research is needed to better understand the role of partner-related attachment styles and prenatal representations during pregnancy, their implications for the postnatal infant–father relationship, and how to intervene during pregnancy to enhance the father-and-child development.

## Figures and Tables

**Table 1 children-10-01187-t001:** Descriptive statistics and correlations between paternal prenatal representations, attachment, and control variables.

*Paternal Prenatal Representation*	1	2	3	4	5	6	7
1. Strongest Feeling about Infant	-						
2. Infant Personality	0.20 **	-					
3. Relationship with Infant	0.24 **	0.27 **	-				
4. ECR Avoidance	0.15 **	0.17 **	0.16 **	-			
5. ECR Anxiety	0.06	−0.06	0.01	0.35 **	-		
6. Parity	0.12 *	0.04	0.15 **	0.03	0.02	-	
7. Education	−0.04	0.04	−0.04	0.01 **	0.01	−0.08	-
N	398	398	398	396	396	396	396
Means	0.06	0.23	0.41	33.20	38.26	0.38	3.04
*SD*1	0.22	4.24	0.49	12.17	14.52	0.07	0.49

Note. *SD* = Standard deviation; N = Number of participants; * *p* < 0.05. ** *p* < 0.01; ECR = Experiences in close relationships.

**Table 2 children-10-01187-t002:** Logistic regression analyses for fathers’ avoidant attachment and prenatal representations.

(Q1) Strongest Feeling about the Child								
							95% CI for EXP (B)	
	B	S.E.	Wald	df	Sig.	Exp (B)	Lower	upper
ECR avoidance	0.59	0.20	7.86	1	0.01	1.80	1.19	2.73
Previous children	1.02	0.47	4.75	1	0.03	2.78	1.11	6.97
Education	−0.13	0.25	0.30	1	0.58	0.87	0.54	1.41
**(Q2) Childs personality**								
							95% CI for EXP (B)	
	B	S.E.	Wald	df	Sig.	Exp (B)	Lower	upper
ECR avoidance	0.39	0.12	10.30	1	0.01	1.48	1.16	1.87
Previous children	0.19	0.25	0.59	1	0.44	1.21	1.2	1.96
Education	0.12	0.14	0.73	1	0.39	1.12	0.86	1.46
**(Q3) Thoughts and feelings about relationship**								
							95% CI for EXP (B)	
	B	S.E.	Wald	df	Sig.	Exp (B)	Lower	upper
ECR avoidance	0.35	0.11	9.82	1	0.02	1.42	1.14	1.75
Previous children	0.62	0.21	8.33	1	0.46	1.85	1.22	2.81
Education	−0.09	0.12	0.56	1	0.42	0.92	0.73	1.15

Note. Father’s education, and parity, are included as covariates in all analyses. ECR avoidance has been z-transformed in order to be more readily interpreted. B = Beta; S.E. = Standard error; df = degrees of freedom; Exp (B) = Expected beta; C.I. = Confidence interval.

## Data Availability

Data is unavailable due to ethical restrictions, but data can be made available upon reasonable request and ethical approval.

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
