# Peer review of "Becoming Dad: Expectant Fathers’ Attachment Style and Prenatal Representations of the Unborn Child"

_children, 2023, doi:10.3390/children10071187_

Round 1
Reviewer 1 Report
I read this paper about expectant fathers and their prenatal representations of their unborn child with great interest. This topic may be relevant to fields such as developmental psychology, family studies, and social work. One of the strengths of this article is that it is the first study to investigate how fathers' partner-related attachment style predicts subdomains of fathers' prenatal representations. Additionally, the article explores the connection between fathers' thoughts and feelings about their unborn child and their parenting behaviors after birth, which can contribute to identifying possible developmental pathways of the postnatal father-child relationship. However, some limitations should be addressed. First, the authors do not provide specific information about common prenatal representations that expectant fathers have of their unborn child. I believe that fathers' prenatal representations of their unborn child may have lasting associations with their subsequent caregiving and the child's development. Understanding what may predict the father-child relationship in the prenatal period can contribute to identifying possible developmental pathways of the postnatal father-child relationship. There is limited knowledge about such antecedents of fathers' prenatal representations, so this study can be considered as a first piece of preliminary evidence. Second, the authors do not provide information about any differences in the prenatal representations and parenting behaviors of first-time fathers versus experienced fathers. The article mainly focuses on exploring the connection between fathers' thoughts and feelings about their unborn child and their parenting behaviors after birth. Third, another limitation of this article includes the sample size which was relatively small, and the study was conducted in a specific cultural context. The authors should thus properly contextualize their work. Additionally, the assessment methods used to assess prenatal paternal representations in this study are not the same as those used in other studies, which may make it difficult to compare results in a precise manner. Finally, there is limited knowledge about antecedents of fathers' prenatal representations, which may limit the generalizability of the findings. The discussion should also elaborate on this.
The article is written in standard academic English with technical terms and jargon commonly used in research articles on this topic.
Author Response
Letter to reviewers
We thank the editor and reviewers for their valuable feedback. We have edited the manuscript according to the suggestions.
We believe this has significantly improved the paper. Our responses to the specific issues raised by the reviewers are as follows:
#Reviewer 1
#1
First, the authors do not provide specific information about common prenatal representations that expectant fathers have of their unborn child. I believe that fathers' prenatal representations of their unborn child may have lasting associations with their subsequent caregiving and the child's development. Understanding what may predict the father-child relationship in the prenatal period can contribute to identifying possible developmental pathways of the postnatal father-child relationship. There is limited knowledge about such antecedents of fathers' prenatal representations, so this study can be considered as a first piece of preliminary evidence.
Answer: Thank you for highlighting this issue. We agree that it would be important to include information about common representations that expectant fathers have of their unborn child. To the best of our knowledge the studies in the field have so far not presented what are common paternal prenatal representations. However, in our manuscript we have now provided more details e.g. referring to how prenatal representations have been measured in previous research, for example by semi-structured interviews and questionnaires. We have provided information of common themes these measures are focusing on (please see page 2-3, line 88-93).
#2
Second, the authors do not provide information about any differences in the prenatal representations and parenting behaviors of first-time fathers versus experienced fathers.
Answer: Thank you for pointing out an important issue. We agree that information about differences in prenatal representations among first-time fathers, and fathers already having children is important to add. We have, added information about the differences in associations between attachment style and prenatal representations among experienced versus non-experienced fathers in our sample in line 310-314 (Results). In addition, we also, discuss other possible explanations of our results and have added additional references in the discussion please see lines 460-473.
#3
Third, another limitation of this article includes the sample size which was relatively small, and the study was conducted in a specific cultural context. The authors should thus properly contextualize their work.
Answer: Thank you for this useful feedback. We agree that both sample size and sample characteristics are important to address. In line 189-191 we have added information about the recruitment of participants. In line 201-203 (Methods) we have added information about the fathers mean age, if the fathers were living with their partner or not, and if the pregnancy was wanted or not. In line 487-489 (Limitations) we have addressed how sample characteristics may limit the generalizability of our findings.
#4
The assessment methods used to assess prenatal paternal representations in this study are not the same as those used in other studies, which may make it difficult to compare results in a precise manner.
Answer: Thank you for pointing out an important issue. On page 11, lines 491-493 we have added this to the limitations of the study and highlighted that this limits the comparability of our findings with other studies on fathers prenatal representations. We have also added in lines 492-493 that the knowledge about prenatal representations is very limited, and this is further highlighted in the limitations/future directions/discussion section.
#5
Finally, there is limited knowledge about antecedents of fathers’ prenatal representations, which may limit the generalizability of the findings. The discussion should also elaborate on this.
Answer: Thank you for this feedback. We have now elaborated on this issue, please see lines 497-498. We have added that the knowledge about prenatal representations is very limited, and we also highlight that there is a need to replicate our findings.
#6
Moderate editing of English language required.
Answer: Thank you for this feedback. We have carefully checked the language for grammatical and spelling mistakes, and we have also carefully revised long sentences to improve its readability.
Reviewer 2 Report
I have reviewed the manuscript "Becoming Dad: Expectant Fathers Attachment Style and Prenatal Representations of the Unborn Child" (children-2433552) and I appreciated the authors' efforts in this study. However, I have identified several areas to address before the manuscript can be considered for publication.
Specifically, Authors should consider the following recommendations:
- I recommend a linguistic revision of the manuscript before resubmitting it to improve its readability.
- The authors should provide a rationale for the sample size and ensure that the sample is appropriately characterized to generalize the results.
Minor editing of English language required
Author Response
#Reviewer 2
#1
I recommend a linguistic revision of the manuscript before resubmitting it to improve its readability.
Answer: Thank you for this feedback. We have carefully checked the manuscript to improve its readability.
#2
The authors should provide a rationale for the sample size and ensure that the sample is appropriately characterized to generalize the results.
Answer: Thank you for highlighting this. We agree that both sample size and sample characteristics are important to address. In the Methods section, line 189-191 we have added information about the recruitment of participants. In line 199-203 we have added information about the fathers mean age, if the fathers were living with their partner or not, and if the pregnancy was wanted or not. In line 487-491 (Limitations) we have addressed how sample characteristics may limit the generalizability of our findings.